# Acceptability of HPV Vaccines: A Qualitative Systematic Review and Meta-Summary

**DOI:** 10.3390/vaccines11091486

**Published:** 2023-09-14

**Authors:** María-Teresa Urrutia, Alejandra-Ximena Araya, Macarena Gajardo, Macarena Chepo, Romina Torres, Andrea Schilling

**Affiliations:** 1School of Nursing, Universidad Andrés Bello, Santiago 8370134, Chile; alejandra.araya.g@unab.cl (A.-X.A.); paz.chepo@unab.cl (M.C.); 2School of Medicine, Universidad de Chile, Santiago 9170022, Chile; mgajardou@gmail.com; 3Sistema de Bibliotecas UC, Pontificia Universidad Católica de Chile, Santiago 8331150, Chile; rtorree@uc.cl; 4Clinical Research Center, Institute of Science and Innovation in Medicine, Facultad de Medicina, Clinica Alemana Universidad del Desarrollo, Santiago 7610315, Chile; aschilling@udd.cl

**Keywords:** acceptability of healthcare, cervical cancer prevention, papillomavirus vaccines

## Abstract

In 2006, the human papillomavirus (HPV) vaccine was approved for use as an effective intervention for reducing the risk of developing cervical cancer; however, its successful implementation is dependent on acceptability. This study aims to provide a comprehensive understanding of the reasons that favor or do not favor the acceptability of HPV vaccines. Methods: We conducted a systematic review and meta-summary of qualitative research on 16 databases. A total of 32 articles that considered the perspectives of vaccine users, their parents, and the professionals who care for them were reviewed. Synthesis was conducted as described by Sandelowski and Barroso. Results: We used inductive and deductive methods to obtain a total of 22 dimensions, out of which three issues stood out that should be considered to improve acceptability and are formed by three groups of study, namely, information about the vaccine, fears and side effects, and sexuality associated with the vaccine. Conclusions: Acceptability, as well as adherence to HPV vaccination, is a complex concept. This review highlights the perspectives of the three sets of actors involved in the process (i.e., users, parents, and professionals) and views these factors in relation to acceptability as a guide for new interventions.

## 1. Introduction

Cervical cancer (CC) is one of the most frequently diagnosed forms of cancer, with around 604,127 new cervical cancer cases diagnosed annually in the world, and is the fourth leading cause of death due to cancer in women, with about 341,831 cervical cancer deaths annually [1,2], Its distribution is uneven around the world, with around 85% of the women with CC living in low- and lower–middle-income countries [1,2]. High-income countries, on the other hand, have been seeing a steady rise in oropharyngeal cancers due to HPV, which mainly affect men and surpass CC in absolute numbers in some cases [2]. Thus, vaccinating female adolescents is the most effective intervention for reducing the risk of developing CC, which also leads to the protection of unvaccinated women, [3]. The global strategy for eliminating CC proposes that 90% of female adolescents should be fully vaccinated with the human papillomavirus (HPV) vaccine by 15 years of age [4]. Depending on the vaccine used, herd protection can also produce a reduction in anogenital warts in women and men [3].

In 2006, the HPV vaccine was approved for use; today, three prophylactic licensed vaccines for the prevention of high-risk HPV infection are available in the majority of countries [5], allowing for gender-neutral vaccination. However, after more than 15 years since its incorporation, the HPV vaccine has failed to achieve the desired coverage [6].

To ensure high levels of acceptance, interventions in HPV vaccination must consider effective strategies for promoting the benefits of the vaccine [4]. Successful implementation is dependent on the acceptability of the intervention to the deliverers and recipients of such an intervention [7]. This study aims to provide a comprehensive understanding of the reasons that favor or do not favor the acceptability of HPV vaccines.

## 2. Materials and Methods

This is a systematic review and meta-summary of qualitative research. The review follows the recommendations of the Preferred Reporting Items for Systematic Reviews and Meta-Analyses (PRISMA) [8] and conducts a qualitative synthesis according to the methodology described by Sandelowski and Barroso [9]. This work aims to answer the following question: What are the aspects that favor or do not favor the acceptability of (and, therefore, adherence to) the HPV vaccine?

### 2.1. Search Strategy

We conducted a systematic search on 16 electronic databases, where the initial search was conducted in March 2019 and updated on February 2020. The search was designed and conducted by the team’s librarian (R.T.), who sought to obtain adequate sensitivity (i.e., a large proportion of relevant studies) and specificity (i.e., a low proportion of unrelated studies). Indexed terms (i.e., Medical Subject Headings) were established and accompanied by free-text keywords to identify a greater number of variants for each concept. The criteria in the search for references in the databases were as follows: published in the period between the years 2006 and 2020, without filtering by language and type of publication. The total number of articles found was 4380 (see Appendix A for the search strategy). Notably, the search strategy was created to answer the aspects of vaccine acceptability from the qualitative point of view and the effectiveness of educational interventions from the quantitative point of view. Although the methodology describes the search strategy, and although the selection of records considered both methodologies, the results focus only on the qualitative research on acceptability.

### 2.2. Inclusion/Exclusion Criteria

All records obtained from the search strategy were imported to the Covidence platform (Veritas Health Innovation, Melbourne, Australia) [10], out of which 1384 were duplicates. These were omitted, which left a total of 2996 articles for the screening phase. Table 1 presents the inclusion criteria considered for selecting the records.

We included only studies that reported primary data. Out of the total records (n = 2996), 22 were reviews (systematic or integrative), which were excluded; however, their references were analyzed to identify any studies that were not covered in the search. We analyzed a total of 279 references, out of which 49 references were included for screening, which left a total of 3045 texts (2996 + 49) for the phase of selection by title and abstract.

### 2.3. Selecting and Reading Primary Research

Two researchers (M.G., M.C., or A.S.) independently evaluated the title and abstract of each record. In the event of disagreement, a third evaluator intervened (A.A. or M.-T.U.). Finally, the researchers selected a total of 550 articles for full-text analysis. The authors of the articles were contacted in cases where the reference lacked full results; out of the 94 contacts, 61 were unsuccessful, which left 489 articles for full-text screening. After reading the full text, we omitted 361 articles, leaving 128 studies that entered the data extraction phase (Figure 1).

### 2.4. Data Extraction and Synthesis of Results

Two researchers (M.G. and M.C.) independently conducted data extraction from the 128 selected studies (quantitative: 98; qualitative: 30). To extract data, a standardized document was created. The extracted data forms were reviewed (M.-T.U. and A.-X.A.), and we obtained 100% congruence between the reviewers. The quality of each article was evaluated against the specific guidelines for JBI qualitative studies [11]; it should be noted that after this evaluation, none of the selected articles should have been eliminated.

As previously cited, this paper will discuss only the findings from the qualitative studies. Out of the 30 qualitative studies [12,13,14,15,16,17,18,19,20,21,22,23,24,25,26,27,28,29,30,31,32,33,34,35,36,37,38,39,40,41], 2 [12,21] presented results in more than one study group; therefore, they were analyzed as 2 separate studies, which generated a total of 32 analyzed studies. Regarding the countries in which the studies were conducted, more than half (n = 17) were conducted in the United States. The groups of respondents were parents/guardians (n = 14; research), healthcare providers (n = 11; research), and people who could or did receive the HPV vaccine (named “vaccinated”; n = 7; research). The sample size varied from 8 to 132. Table 2 presents the characteristics of each study.

We conducted a synthesis according to the guidelines proposed by Sandelowski and Barroso [9] and used the theoretical framework of acceptability described by Sekhon [7] to organize the findings. Acceptability is defined as a “multi-faceted construct that reflects the extent to which people delivering or receiving a healthcare intervention consider it to be appropriate, based on anticipated or experienced cognitive and emotional responses to the intervention.” Moreover, the framework has seven components, namely, affective attitude, burden, perceived effectiveness, ethicality, intervention coherence, opportunity cost, and self-efficacy [7].

## 3. Results

We conducted the synthesis of the results according to the five stages described by Sandelowski and Barroso [9]. A diagram of the synthesis process is presented in Figure 2.

### 3.1. Extraction of Findings: “This Stage Entails Distinguishing the Specific Finding You Want to Integrate from All Other Elements in the Research Reports Containing Those Finding” [9]

The first step was to define what type of findings would be considered: findings were defined as “any researcher description addressing the opinion or experience from the people about vaccine acceptability supported by an original quotation”. The second step was to define the section from which the findings would be extracted: they would only be extracted from Results sections.

A total of 842 findings were extracted from the 32 studies, and all of them were incorporated into MAXQDA software (Verbi Software, Berlin, Germany); the name of the original dimension/category described per article was preserved. Afterward, the original description that the author made for each dimension/category was transferred as a *memo*, which was used at the time of the synthesis.

### 3.2. Editing Findings: “Once You Have Finished Extracting Findings, You Should Edit Them to Make Them Accessible as Possible to Any Reader” [9]

In order to preserve the original dimensions/categories’ description, and given that the findings were understandable without editing, edition was not necessary. Therefore, descriptions and stories were left unchanged.

### 3.3. Grouping Findings: “To Group Findings That Appear to Be the Same Topic” [9]

Using the deductive method, we organized the findings from each study according to the components of the theoretical framework of acceptability [7]. The grouping was performed in each study group (parents, vaccinated, and healthcare providers): the author of the theoretical framework’s definition of the seven components was the guide for classifying each finding. Two researchers (M.-T.U. and A.-X.A.) independently performed the grouping until a consensus was reached throughout the process. Therefore the 842 findings were grouped in the six components of the framework, and the self-efficacy component was the only one left unmatched (Table 3).

### 3.4. Abstraction of the Findings: “In the Abstraction Process, You Will Further Reduce the Many Statements of Findings You Extracted, Edited, and Grouped into More Parsimonious Rendering of Them” [9]

After grouping the findings, the two researchers independently conducted the abstraction of the findings until a consensus was reached. This stage was performed in two phases using the inductive method, as follows:

#### 3.4.1. Phase 1: Generating New Categories

For each study group, in each component of the theoretical model, new categories supported by the original quotations were generated. In this phase, the definitions of each dimension of the theoretical model were preserved as a guide. Therefore, in the vaccinated group, 19 new categories were generated, supported by 142 original quotations; in the group of fathers/mothers, there were 41 new categories supported by 310 original quotations; and in the group of professionals, there were 59 new categories supported by 390 quotations (see Table 3).

#### 3.4.2. Phase 2: Generating Dimensions from the New Categories

After identifying the new categories in phase 1, dimensions were created, defined as “a group of categories whose meanings were similar”; in this phase, the grouping was guided by the meaning of the category, and not by the component of the theoretical model to which it belonged (Table 4). The analysis revealed 22 dimensions, all of which considered the aspects that favor or do not favor the acceptability of HPV vaccines. Specifically, we identified nine, nine, and four dimensions for the healthcare providers, parents/guardians, and vaccinated individuals, respectively. The lower number of dimensions generated in the vaccinated group can be explained by the lower number of articles analyzed in this group. It should be noted that all of the dimensions created came from categories belonging to different components of the theoretical frameworks used previously.

### 3.5. Calculating Manifest Frequency and Intensity Effect Sizes: “The Calculation of Effect Sizes Constitutes a Quantitative Transformation of Qualitative Data in the Service of Extracting More Meaning from Those Data and Verifying the Presence of a Pattern or Theme” [9]

The fifth step consisted of the generation of the inter-study matrix, which enabled the organization of the articles according to the identified dimensions, and the intra-study matrix, which promoted the classification of the dimensions cited by each study. The first pinpointed the observed frequency, and the second highlighted the observed intensity of effect sizes. Both matrices were generated for each study group (Table 5).

The inter-study matrix indicated that the dimensions generated were supported by a significant number of studies, whose lowest frequency was 27%. It should be noted that five dimensions had a frequency of more than 90%. In relation to the intra-study matrix, the range between 22% and 100% revealed that all of the studies included in this research support the generated dimensions.

Based on this last stage, and to answer the research question, we carried out a selection process of those dimensions that were relevant in the three study groups. The first step was to select the dimensions with a frequency >60% per group: of the 22 dimensions, 13 dimensions were retained. The second step was to select those dimensions with similar topics between groups; therefore, nine dimensions were selected (three per group).

The three themes whose meanings were shared by the three study groups were information about the vaccine (see Table 4) (dimensions number two in the healthcare providers group, number one in the parents/guardians group, and number one in the vaccinated group), fears and side effects of the vaccine (dimensions number six in the healthcare providers group, number three in the parents/guardians group, and number two in the vaccinated group), and aspects of sexuality associated with the vaccine (dimensions number four in the healthcare providers group, number four in the parents/guardians group, and number three in the vaccinated group). Table 6 presents the themes, their definitions, and quotations per group.

## 4. Discussion

CC is undoubtedly the paradigm of health inequity, given that more than 85% of individuals suffering from it are undereducated women who live in the world’s poorest communities, who are located not only in low- and lower–middle-income countries but also in underserved areas/minorities of higher–middle- and high-income countries [2,4]. Given the relevance of this disease, any effort made in order to answer the World Health Organization’s call must be considered; for this reason, this qualitative study aims to increase vaccine coverage by understanding the underlying experiences, perceptions, and behavior more properly though a systematic review and meta-summary.

The main strength of this study is that it analyzes the acceptability of HPV vaccines from three perspectives, namely, those who receive the vaccine, those who authorize it (parents/guardians), and those who recommend it (healthcare providers), as the existing reviews on the subject were carried out in only one study group [42,43,44,45,46]. The second strength is that, to the best of our knowledge, no publications have synthesized qualitative research that addresses the issue of the acceptability of HPV vaccines. The selected reviews in this study address barriers [47], decision-making processes [48,49], interventions [50], and messages in social networks [51], among others. Notably, a review was published in 2020 [52] that analyzed the social aspects related to the acceptance of the vaccine. However, said publication only addresses the topic within the context of Colombia. Other interesting systematic reviews on acceptability and adherence have been published, but only with quantitative findings [53,54] or in other study groups [55]. The third strength of this review is the number of databases consulted, which ensured broad coverage. The principal limitation of this paper is that the evidence after 2020 was not considered; therefore, the results are related to the dates included. The possible bias of this study is the theoretical lens used in the first part of phase 1 of the analysis, since the findings were grouped according to the categories of the selected theoretical framework; however, this limitation was balanced in the second part of the same phase where, through an inductive process, new categories were created.

Despite being recognized as one of the most successful public health measures, the lack of confidence in vaccines is now considered to be a threat to the success of vaccination programs [56]. Acceptability in healthcare is a complex concept because it reflects interactions between patients and healthcare systems, which include healthcare providers and policymakers [57]. Therefore, the approaches to be used must consider multiple factors. This systematic review demonstrates this complexity; however, it also sheds light on the major components that must be addressed, which are shared by the three groups of respondents.

Information regarding the vaccine is an issue that must be addressed in an urgent manner. It is known that the acceptability of vaccination increases when recommended by healthcare providers [42,45,46,58], such that their recommendations about whether or not to use it may restrict access to the vaccine [49,59]. Therefore, receiving the necessary information is mandatory for healthcare providers to enable them to provide correct prescriptions and to receive training in appropriately addressing this issue. This review highlights the need for professionals to demand this information. To ensure high levels of acceptance and sustained coverage, HPV vaccination programs must be accompanied by strong communication strategies [4]. The WHO recommends investment in communication strategies for the introduction of HPV vaccines [60]. Moreover, the levels of knowledge about HPV vaccines among the population are low [45,61], which has been also associated with their acceptability [62]. However, if professionals lack commitment to vaccination and the tools for delivering accurate information, the low levels of knowledge among the population will not be improved.

Scholars have recognized the relevance of good communication among healthcare professions and communities, which enables adequate sharing of information regarding the risks and benefits of vaccines [52]. In many countries, HPV vaccines have been subject to rumors and linked with adverse events [60]. It is known that fear of adverse effects [46,63] and exposure to rumors are the main reasons for non-vaccination [63]. Moreover, uncertainty about the effectiveness of the vaccines [46], perceptions that vaccination is unnecessary, and concerns regarding the safety and side effects of the vaccines [43] are all barriers to the successful implementation of HPV vaccination. These reasons may be applicable to all vaccines; however, HPV vaccines seemingly present greater concerns, because they are vaccines with *unique characteristics* [60].

These characteristics are related to the unequivocal association between HPV vaccines and sexually transmitted disease, which may be the most important point to address and may increase the significance of the other two topics identified by this review. Acceptability is a term that encompasses the social and cultural factors of healthcare [57]; therefore, cultural values, as well as the dissemination of the HPV vaccines from the gender perspective, are aspects that must be considered.

The link between sexual intercourse and HPV vaccines frequently complicates people’s decisions regarding vaccination [64]. Scholars have reported that concerns that HPV vaccination will promote sexual behavior among adolescents are a barrier to adherence [58], and social norms and values related to sexual activity form the basis of the decision to vaccinate [49]. Parents may decide not to allow their daughters to be vaccinated on the basis of their cultural or religious perceptions about sexual activity [49]. As such, cultural values are an important aspect related to health and healthcare decisions [65,66]. Thus, reinforcing the importance of HPV immunization by providing accurate information in a timely manner, while considering cultural and religious sensitivities and varying levels of health literacy, is crucial [64]. Also, to promote inclusion, policymakers, health professionals, and patient organizations must work together to ensure that the voices of families, parents, and adolescents are included in implementing HPV vaccination strategies.

This review highlights three important factors to consider when addressing HPV vaccines’ acceptability, and all of them can be scaled to different levels, as presented in a previously published review [67]: information about the vaccine is a factor located at the individual and societal levels, because it is related to the lack of awareness among healthcare providers and health literacy among the population; the fears and side effects are also located at the individual and societal levels, and the aspects of sexuality represent a factor that transcends the individual level, since it includes the cultural and religious beliefs of a given community.

As of 2020, less than 30% and 25% of lower–middle- and low-income countries, respectively, have introduced HPV vaccines into their national immunization schedules, in contrast with 85% of high-income countries [4]. The WHO’s 2022 updated vaccination schedule recommended an “alternative, off-label single-dose schedule” of HPV vaccines. The WHO currently recommends a one- or two-dose schedule for girls aged 9–14 years, a one- or two-dose schedule for girls and women aged 15–20 years, and two doses with a six-month interval for women older than 21 years [68]. This new recommendation may help to reduce this large inequity, but it needs to be carefully monitored: “immunization should be a social norm, wherein the demand for and access to immunization for all members of every community is normal, socially acceptable health behavior” [60].

Based on our findings, we suggest that future research should address aspects related to perceived barriers and facilitators regarding the acceptability of the HPV vaccine, considering the diversity of cultural, ethnic, geographic, gender, and migrant contexts, as well as the difference between urban and rural settings. These studies could allow for the development of more appropriate intervention strategies adapted to each sociocultural context. In addition, it is relevant to explore the influence of cultural beliefs and norms in decision-making related to vaccination, especially regarding adverse events. It would also be essential to examine how information on these issues is communicated and exchanged on social networks, which could contribute to the design of more effective promotional strategies. It is essential to incorporate studies that analyze the perceptions and attitudes of health professionals concerning the strategies that they use to recommend the HPV vaccine. This strategy would make it possible to identify areas for improvement in communication with patients and their families, facilitating a more comprehensive and appropriate approach to the individual context of each patient. Finally, it is important to consider studies that include the experiences and perceptions of teachers, since in many cases they are the ones who must face the questions of parents and students about the vaccine.

## 5. Conclusions

The acceptability of and adherence to HPV vaccines result from an intricate interaction between social and cultural factors related to sexuality, knowledge and understanding of the information received about vaccines, and fears related to their adverse effects. This interaction becomes relevant when considering the perspectives of those who receive the vaccine (adolescents), those who authorize it (parents), and those who promote and administer it (health professionals).

The three main dimensions generated from the analysis guide the conclusions of this article. Health information is often discussed; however, if it continues to be a concern for patients and health professionals, it is because it is still a weakness that should be improved (dimension one of this review). It is vitally important that the information provided is accurate and capable of allaying fears associated with HPV vaccines, since this topic was relevant in the three study groups in this article (dimension two of this review). Likewise, it is crucial to consider the cultural aspects related to these vaccines to achieve greater acceptability among the population. Talking about sexuality (dimension three of this review) is a sensitive topic and, therefore, must be treated with the best of cultural skills. Through a more informed and culturally sensitive approach, we will be able to foster greater acceptance of HPV vaccines and, ultimately, contribute to the overall health and protection of the population.

Finally, it is important to highlight the value of information from qualitative studies. This review was conducted using a scientific methodology described for the synthesis of qualitative research, and as said by its author, “qualitative research synthesis—by itself—constitutes a form of scientific inquiry” [9]. The experience and opinion of the protagonists in relation to the HPV vaccine is one more source of evidence to carry out correct decision-making and, therefore, improve our clinical practice. Technological progress in HPV vaccines will never be useful if those aspects that reduce their acceptability are not considered.

## Figures and Tables

**Figure 1 vaccines-11-01486-f001:**
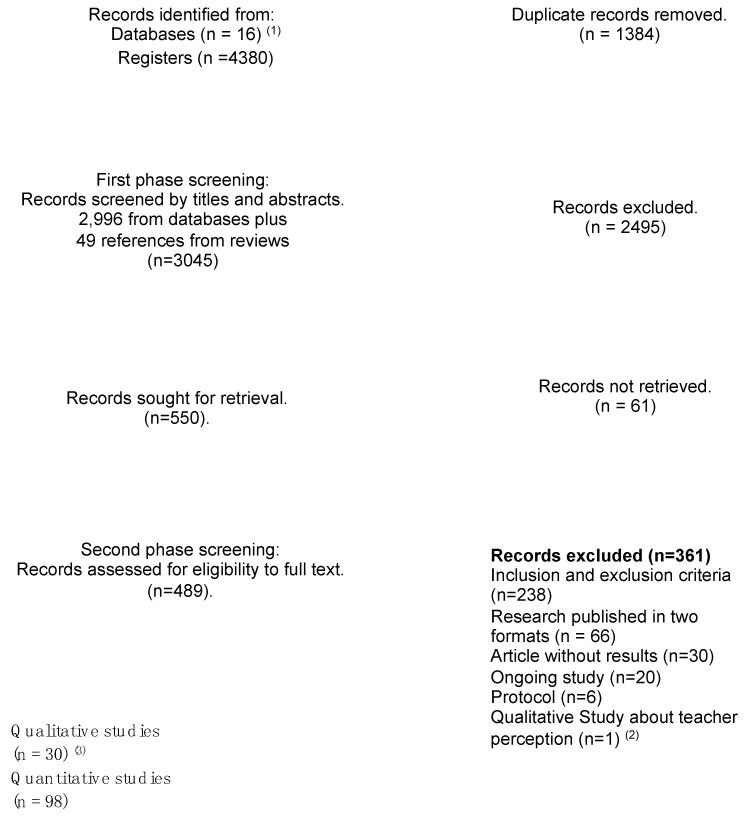
Flowchart of the record selection process. ^(1)^ See Appendix A for information about the databases. ^(2)^ This study was excluded because it was the only one whose study group was teachers. ^(3)^ Articles included in this paper.

**Figure 2 vaccines-11-01486-f002:**
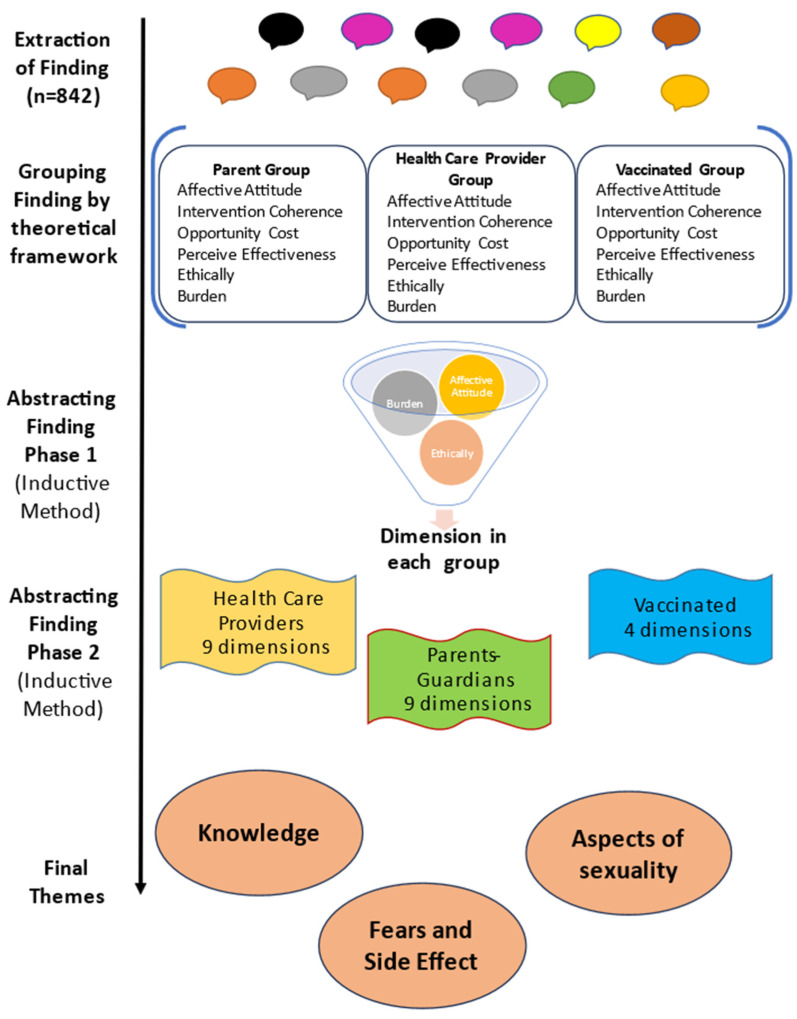
Diagram of the synthesis process.

**Table 1 vaccines-11-01486-t001:** Criteria for record selection.

Criteria	Description
Study aim	Adherence to and/or acceptability of the vaccine
Type of study	- Quantitative ^1^ - Qualitative - Mixed methods
Study group	- People who receive the vaccine (children–adolescents or adults) - Parents or guardians of the people who receive the vaccine - Teachers at schools where the vaccine is administered - Health professionals who have some kind of participation in the vaccination process
Population	Studies conducted on healthy population with a normal risk of contracting HPV
Type of analysis ^2^	- Separate study groups - Coded “in vivo” - Thematic, conceptual, or interpretative analysis ^3^

^1^ In the case of quantitative studies that contain an intervention. ^2^ Applicable only to qualitative studies. ^3^ According to the description by Sandelowski and Barroso [8].

**Table 2 vaccines-11-01486-t002:** Summary of the included studies (n = 32).

First Author (Ref)	Group of Study	Country	Sample Description	Ages of the Samples (Years)	Gender	Data Collection Method
Perkins [12]	Parents	United States	Parents of girls with two or three doses of vaccine (n = 65)	41 to 49	Both	Interview
Dempsey [13]	Parents	United States	Mothers of adolescents aged between 11 and 17 years (n = 52)	40 to 44	Women	Telephone interview
Gottvall [14]	Parents	Sweden	Parents who allowed the vaccination of their daughters aged between 11 and 12 years (n = 29)	44	Both	Interview
Blumbling [15]	Parents	United States	Parents over the age of 18 years with a child between the ages of 9 and 13 (n = 18)	35 to 60	Both	Focus group
Roncancio [16]	Parents	United States	Latina mothers of boys and girls between the ages of 11 and 17 years (n = 51)	42	Women	Interview
Gottvall [17]	Parents	Sweden	Parents of girls aged between 11 and 12 years (n = 42)	Average: 43	Both	Interview
Grandahl [18]	Parents	Sweden	Parents who refused to vaccinate their daughters (n = 23)	Average: 44	Both	Interview
Pitts [19]	Parents	United States	Parents/caregivers of girls aged 9–13 years (n = 33)	No information	Both	Focus group
Warner [20]	Parents	United States	Parents of Latino boys/girls aged 11–17 years (n = 52)	18 to 50	Both	Focus group
Fernández [21]	Parents	Puerto Rico	Mothers of daughters aged between 16 and 26 years (n = 30)	Average: 47.9	Women	Focus group
Marlow [22]	Parents	United Kingdom	Mothers with at least one daughter aged less than 16 years (n = 20)	No information	Women	Interview
Craciun [23]	Parents	Romania	Mothers of girls in the vaccine target group (n = 25)	30 to 50	Women	Focus group and interview
Roncancio [24]	Parents	United States	Spanish-speaking Hispanic mothers of adolescent girls and boys aged 11–17 years (n = 85)	Average: 39	Women	Interview
Btoush [25]	Parents	United States	Latina mothers of HPV-vaccine-eligible children (n = 132)	40 (50%)	Women	Focus group
Perkins, 2016 [12]	Health care providers	United States	Healthcare providers (n = 33)	No information	No information	Interview
Mazza [26]	Health care providers	Australia	General practitioners (n = 24)	34 to 75 Average: 49	Both	Telephone interview
Rubens–Augustson [27]	Health care providers	Canada	Family physicians (n = 8), nurse practitioners (n = 2), and a gynecologist (n = 1)	18 to 56	Both	Interview
Head [28]	Health care providers	United States	Nurse practitioners (n = 3), licensed practical nurses (n = 4), and a medical doctor (n = 1)	Average: 38.8	Women	Interview
Rockliffe [29]	Health care providers	United Kingdom	Healthcare providers (n = 28)	No information	Both	Focus group
Cartmell [30]	Health care providers	United States	State leaders with the potential to influence vaccination policies and practices (n = 34)	45 to 64	Both	Interview
Carhart [31]	Health care providers	United States	Stakeholders involved with aspects of care directly related to HPV vaccination, policy, industry, research, or cancer outreach/community engagement (n = 31)	No information	Both	Interview
Ayele [32]	Health care providers	United States	Healthcare providers (n = 26)	No information	Both	Interview
Ng [33]	Health care providers	United States	Health plan directors (n = 10)	No information	No information	Interview
Javanbakht [34]	Health care providers	United States	Physicians (n = 4), a physician’s assistant (n = 1), medical assistants (n = 7), and case managers (n = 9)	No information	Both	Interview
Lefevre [35]	Health care providers	France	Physicians (n = 16)	No information	No information	Interview
Fernández [21]	Vaccinated	Puerto Rico	Women aged between 16 and 26 years (n = 30)	Average: 20.4	Women	Focus group
Oscarsson [36]	Vaccinated	Sweden	Women (n = 16)	17 to 26	Women	Interview
Miller [37]	Vaccinated	United States	Adolescents (n = 50)	14 to 18	Both	Focus group
Gao [38]	Vaccinated	United States	People aged between 18 and 34 years (n = 44)	Average: 24.6	Both	Focus group
Carnegie [39]	Vaccinated	Scotland	People aged between 16 and 26 years (n = 40)	16 to 26	Both	Focus group and interview
Siu [40]	Vaccinated	Hong Kong	Undergraduate Chinese students (n = 35)	19 to 23	Women	Interview
Lim [41]	Vaccinated	Singapore	Female students (n = 40)	18 to 26	Women	Focus group and interview

**Table 3 vaccines-11-01486-t003:** Findings from each study group, classified according to the theoretical framework of acceptability (grouping findings) and new categories that emerged from abstracting the findings (phase 1).

	Groups under Study	AA	B	E	IC	OC	PE	SE	Total
Grouping findings	Healthcare providers	76	106	55	97	18	38	0	390
	Parents	47	88	27	91	6	51	0	310
	Vaccinated individuals	19	16	27	60	3	17	0	142
		142	210	109	248	27	106	0	842
Abstracting findings (phase 1)	Healthcare providers	18	15	6	11	5	8	--	63
	Parents	11	9	7	7	2	9	--	45
	Vaccinated individuals	3	3	4	5	1	3	--	19
		32	27	17	23	8	20	--	127

AA = affective attitude, B = burden, E = ethicality, IC = intervention coherence, OC = opportunity cost, PE = perceived effectiveness, SE = self-efficacy.

**Table 4 vaccines-11-01486-t004:** New dimensions for each group under study.

Groups under Study	Dimensions	Components of the Model to Which the Categories Belong
Healthcare providers	1. Cost and public policies related to vaccines	7 components: AA-B-E-IC-OC-PE
	2. Vaccine information and education	4 components: AA-B-IC-OC
	3. Lack of time/other priorities	5 components: AA-B-IC-OC-PE
	4. Associated between vaccination and sexuality	2 components: E-IC
	5. Record and reminder systems	4 components: B-IC-OC-PE
	6. Vaccine safety/fears	3 components: AA-B-PE
	7. Strategies for promoting vaccination	4 components: AA-B-IC-PE
	8. Vaccine mandatory/decision/doses	4 components: AA-B-E-PE
	9. Cultural and language differences	3 components: AA-B-E
Parents/guardians	1. Information is needed	5 components: AA-B-E-IC-PE
	2. The vaccine is beneficial and necessary	3 components: AA-IC-PE
	3. The vaccine may cause harm/side effects	3 components: AA-B-PE
	4. Vaccination is associated with sexuality and gender roles	5 components: AA-B-E-IC-PE
	5. The vaccine is mandatory (trust or mistrust)	4 components: AA-E-PE
	6. Vaccine cost and access	2 components: B-OC
	7. Decision to vaccinate	4 components: AA-B-IC-PE
	8. Age upon vaccination	2 components: AA-IC
	9. Reminders to vaccinate	3 components: B-IC-PE
Vaccinated individuals	1. Knowledge about vaccines	5 components: AA-B-E-IC-PE
	2. Risk perception and associated fears	4 components: AA-B-E-PE
	3. Vaccination is associated with sexuality and gender roles	3 components: E-IC-PE
	4. Cost and number of doses of vaccines	2 components: B-OC

AA = affective attitude, B = burden, E = ethicality, IC = intervention coherence, OC = opportunity cost, PE = perceived effectiveness, SE = self-efficacy.

**Table 5 vaccines-11-01486-t005:** Inter- and intra-study matrices.

	Categories																						
	Healthcare Providers									Parents/Guardians									Vaccinated Individuals				
First Author, Year (Cite)	1	2	3	4	5	6	7	8	9	1	2	3	4	5	6	7	8	9	1	2	3	4	Intensity Effect Size
Perkins, 2016 [12]	-	-	-	-	-	-	-	-	-	Yes	Yes	No	No	No	Yes	Yes	No	Yes	-	-	-	-	56%
Dempsey, 2009 [13]	-	-	-	-	-	-	-	-	-	No	No	Yes	Yes	No	No	Yes	No	No	-	-	-	-	33%
Gottvall, 2013 [14]	-	-	-	-	-	-	-	-	-	Yes	Yes	Yes	No	Yes	No	Yes	Yes	No	-	-	-	-	67%
Blumling, 2014 [15]	-	-	-	-	-	-	-	-	-	No	No	Yes	Yes	No	No	No	No	No	-	-	-	-	22%
Roncancio, 2017 [16]	-	-	-	-	-	-	-	-	-	Yes	Yes	Yes	No	Yes	Yes	Yes	No	Yes	-		-	-	78%
Gottvall, 2017 [17]	-	-	-	-	-	-	-	-	-	Yes	No	No	Yes	Yes	No	No	No	No	-	-	-	-	33%
Grandahl, 2014 [18]	-	-	-	-	-	-	-	-	-	Yes	No	Yes	Yes	Yes	No	No	Yes	No	-	-	-	-	56%
Pitts, 2013 [19]	-	-	-	-	-	-	-	-	-	Yes	Yes	Yes	Yes	Yes	No	Yes	Yes	No	-	-	-	-	78%
Warner, 2015 [20]	-	-	-	-	-	-	-	-	-	Yes	No	Yes	Yes	No	Yes	No	No	No	-	-	-	-	44%
Fernández, 2014 [21]	-	-	-	-	-	-	-	-	-	Yes	No	Yes	Yes	No	Yes	Yes	No	No	-	-	-	-	56%
Marlow, 2009 [22]	-	-	-	-	-	-	-	-	-	Yes	Yes	Yes	Yes	Yes	No	Yes	No	No	-	-	-	-	67%
Craciun, 2012 [23]	-	-	-	-	-	-	-	-	-	Yes	No	Yes	No	Yes	Yes	Yes	No	No	-	-	-	-	56%
Roncancio, 2019 [24]	-	-	-	-	-	-	-	-	-	Yes	Yes	Yes	No	No	Yes	Yes	No	No	-	-	-	-	56%
Btoush, 2019 [25]	-	-	-	-	-	-	-	-	-	Yes	No	No	Yes	No	No	Yes	Yes	Yes	-	-	-	-	44%
Perkins, 2016 [12]	No	No	No	No	Yes	No	No	Yes	No	-	-	-	-	-	-	-	-	-	-	-	-	-	22%
Mazza, 2014 [26]	Yes	Yes	Yes	No	No	Yes	Yes	Yes	No	-	-	-	-	-	-	-	-	-	-	-	-	-	77%
Rubens- Augustson, 2019 [27]	Yes	Yes	Yes	Yes	No	No	Yes	Yes	Yes	-	-	-	-	-	-	-	-	-	-	-	-	-	77%
Head, 2013 [28]	No	Yes	No	Yes	Yes	Yes	Yes	Yes	No	-	-	-	-	-	-	-	-	-	-	-	-	-	66%
Rockliffe, 2020 [29]	No	Yes	No	No	No	Yes	Yes	Yes	No	-	-	-	-	-	-	-	-	-	-	-	-	-	44%
Catmell, 2018 [30]	Yes	Yes	Yes	Yes	Yes	Yes	Yes	No	Yes	-	-	-	-	-	-	-	-	-	-	-	-	-	89%
Carhart, 2018 [31]	Yes	Yes	Yes	Yes	Yes	Yes	Yes	Yes	Yes	-	-	-	-	-	-	-	-	-	-	-	-	-	100%
Ayele, 2018 [32]	Yes	Yes	Yes	Yes	No	No	Yes	Yes	No	-	-	-	-	-	-	-	-	-	-	-	-	-	55%
Ng, 2017 [33]	No	Yes	No	Yes	Yes	No	Yes	Yes	No	-	-	-	-	-	-	-	-	-	-	-	-	-	55%
Javanbakht, 2012 [34]	Yes	Yes	No	Yes	Yes	Yes	Yes	Yes	No	-	-	-	-	-	-	-	-	-	-	-	-	-	77%
Lefevre, 2018 [35]	Yes	Yes	Yes	Yes	No	Yes	Yes	Yes	No	-	-	-	-	-	-	-	-	-	-	-	-	-	77%
Oscarson, 2012 [36]	-	-	-	-	-	-	-	-	-	-	-	-	-	-	-	-	-	-	Yes	Yes	Yes	Yes	100%
Miller, 2014 [37]	-	-	-	-	-	-	-	-	-	-	-	-	-	-	-	-	-	-	Yes	Yes	No	No	50%
Fernández, 2014 [21]	-	-	-	-	-	-	-	-	-	-	-	-	-	-	-	-	-	-	Yes	Yes	Yes	Yes	100%
Gao, 2016 [38]	-	-	-	-	-	-	-	-	-	-	-	-	-	-	-	-	-	-	Yes	Yes	Yes	No	75%
Carnegie, 2017 [39]	-	-	-	-	-	-	-	-	-	-	-	-	-	-	-	-	-	-	Yes	Yes	No	No	50%
Siu, 2013 [40]	-	-	-	-	-	-	-	-	-	-	-	-	-	-	-	-	-	-	Yes	Yes	Yes	Yes	100%
Lim, 2019 [41]	-	-	-	-	-	-	-	-	-	-	-	-	-	-	-	-	-	-	Yes	Yes	Yes	Yes	100%
Frequency effect size	64%	91%	55%	73%	55%	64%	91%	91%	27%	86%	43%	79%	64%	50%	43%	71%	29%	21%	100%	100%	71%	57%	

**Table 6 vaccines-11-01486-t006:** Themes, definitions, and narrative.

Knowledge	This theme reveals the lack of information regarding HPV vaccination in the three groups of respondents and their interest in this regard. Such information enables users to make informed decisions and professionals to prescribe the vaccines. Parents and vaccinated individuals indicate that they lack understanding of the information given to them. In turn, professionals report the need for patients to receive education, and that the information must be consistent. Without truthful information, discussion of and, therefore, adherence to the vaccine become difficult.	
	Healthcare providers	“You never hear a patient say, I think I have HPV … since the research started, we became more knowledgeable and more apt to talk to them about [HPV].” [28]
	Parents/ guardians	“We haven’t received any explanation … no information about HPV has been given. The only thing we got was a vaccination appointment.” [18]
	Vaccinated individuals	“I have heard about it [HPV], but I don’t know what it is.” [21]
Fears and Side effect	This theme unveils all aspects associated with perceived barriers to receiving vaccination and the caution that people exercise in relation to the safety and effectiveness of vaccines. Side effects are part of this theme, as is the risk perception of vaccinated individuals	
	Healthcare professionals	“From the other side there are questions: How old is the vaccine? Is this a clinical trial? Are we guinea pigs?” [30]
	Parents/ guardians	“I just don’t know enough about it. That’s reason number one and then I don’t want her to fall into a category where she gets this done and then ten years down the line they find that it reacts a different way. So it’s a little bit frightening for me.” [13]
	Vaccinated	“What chemicals are they putting inside the HPV shot … How can we trust it?” [37]
Aspects of sexuality	The relationship between the vaccine and sexuality reveals multiple aspects, such as the risk of promoting sexual activity and even promiscuity, because they would be reducing the risk of contracting sexually transmitted diseases; the need to talk about sexuality and the difficulty associated with this discussion; the age group at which the vaccine is directed; the perception that it is a vaccine only for women; and religious aspects considered by a number of participants	
	Healthcare providers	“Well, their concern is the same. They tell you over and over the same, ‘I don’t want my child to have sex, so I don’t want to give the vaccine to my girl ‘cause she’s gonna start having sex.’ And the same, sex, sex, sex.” [34]
	Parents/ guardians	“I think [the HPV vaccine] is important, but I have to inform my son why I’m giving him the vaccine. Sex education is very important, but sometimes as a Hispanic parent we try to avoid those issues ….” [20]
	Vaccinated individuals	“I won’t get the vaccine, because I am not [having sex]. Thus, I am not going to get the vaccine. [I will get the] injection before I get married, together with the premarital checkup.” [38]

## Data Availability

Not applicable.

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
