# Peer review of "Acceptability of HPV Vaccines: A Qualitative Systematic Review and Meta-Summary"

_vaccines, 2023, doi:10.3390/vaccines11091486_

Round 1

Reviewer 1 Report

The manuscript by Urrutia et al. reported a qualitative systematic review of the acceptability of HPV vaccines, highlighting the perspectives from users, patients, and professionals. Below are my comments: 

1. What is the reason to distinguish qualitative and quantitative studies? and what is the rationale to focus on qualitative studies in the current review?

2. As qualitative and quantitative results are usually both presented in one study. What is the criteria to distinguish between quantitative and qualiotative studies?

3. The conclusions need to be improved. It is not clear what conclusions or suggestions can be made from the review performed by the authors. 

4. Flowchart in Fig. 1 is missing.

Author Response

Reviewer 1: The manuscript by Urrutia et al. reported a qualitative systematic review of the acceptability of HPV vaccines, highlighting the perspectives from users, patients, and professionals. Below are my comments: 

  1. What is the reason to distinguish qualitative and quantitative studies? and what is the rationale to focus on qualitative studies in the current review?

Response 1: The reason to distinguish qualitative and quantitative studies was that the aims from each one is different. The qualitative studies were about HPV vaccine acceptability and the quantitative studies were about effective educative interventions to increase the adherence. Because both aims are different and they are not comparable, this paper focus on acceptability.

  1. As qualitative and quantitative results are usually both presented in one study. What is the criteria to distinguish between quantitative and qualiotative studies?

Response 2: We are not talking about type of result; we are talking about type of study. The qualitative and quantitative are different methodologies, therefore the criterion was the type of methodology described by the author in each research.

  1. The conclusions need to be improved. It is not clear what conclusions or suggestions can be made from the review performed by the authors. 

Response 3: Suggestions from the results were added and numbered at the end of the discussion. The suggestion from the reviewer was considered, and the conclusion was improved.

  1. Flowchart in Fig. 1 is missing.

Response 4: We don’t understand what happened, because the flowchart is in page 3 of the paper. In the new version the flowchart was sent as document

Reviewer 2 Report

This is review manuscript may be interested to the public, but lack some scientific merits.

Honestly, I can hardly find novelty from this review paper, and to some extesnsion, I consider this manuscript would meet humanity journals other than scientific journals, like vaccines.

Author Response

Reviewer 2: This is review manuscript may be interested to the public, but lack some scientific merits. Honestly, I can hardly find novelty from this review paper, and to some extesnsion, I consider this manuscript would meet humanity journals other than scientific journals, like vaccines.

Response to reviewer 2: The reason to send this paper was that answer the call of this special number about HPV vaccine and the aims of this journal. The scientific merit is related to methodology, and the novel is described in the discussion section.

Reviewer 3 Report

In the current Review titled “Acceptability of HPV Vaccines: A Qualitative Systematic Review and Meta-Summary”, María-Teresa, al et aimed to provide a comprehensive understanding of the reasons that favor or do not favor the acceptability of the HPV vaccines. The design is suitable and reasonable; the analysis is reliable; the conclusion is informative.

A minor concern is that economic burden as a potential factor is not considered and included.

Author Response

Reviewer 3: In the current Review titled “Acceptability of HPV Vaccines: A Qualitative Systematic Review and Meta-Summary”, María-Teresa, al et aimed to provide a comprehensive understanding of the reasons that favor or do not favor the acceptability of the HPV vaccines. The design is suitable and reasonable; the analysis is reliable; the conclusion is informative. A minor concern is that economic burden as a potential factor is not considered and included.

Response to reviewer 3: The economic burden was in each group of study: number 1 in health care provider group, number 6 in parent/guardians’ group, and number 4 in vaccinated individuals’ group (see table 4). But the effect size frequency was lower than the other topic, therefore was not included in the las step of the analysis. 

Reviewer 4 Report

The study conducted an extensive analysis to explore the acceptability of the human papillomavirus (HPV) vaccine. Through a systematic review and meta-summary of qualitative research, the viewpoints of vaccine users, parents, and healthcare professionals were examined. A total of 32 articles were thoroughly analyzed. The research revealed three critical factors that significantly impact the acceptance of the vaccine: the availability of comprehensive information about the vaccine, concerns and apprehensions regarding potential side effects, and the association of the vaccine with sexuality. This review highlights the complex nature of acceptability and adherence to HPV vaccination, emphasizing the importance of understanding the perspectives of users, parents, and professionals for future interventions. While the study provided valuable insights, further revisions are suggested to improve the clarity and refinement of the manuscript.

1.     It is important to include the most recent epidemiological data available on HPV-related malignancies, as well as benign lesions such as skin warts.

2.     Previous studies have explored the limitations of HPV vaccination accessibility and adherence. To differentiate this study from previously published literature, it is essential to refer to the following papers with their DOIs: 10.1093/ije/dyt049, 10.1016/j.vaccine.2020.07.055, 10.1016/j.eclinm.2021.100836.

3.     There seems to be an issue with Figure 1. Please provide a detailed schematic diagram that illustrates the results of the search strategy.

4.     Why was the search strategy limited only to February 2020? It is crucial to provide a rationale for this choice and consider including published literature beyond that timeframe to ensure the data reflects the current state of HPV vaccine uptake. Information from three years ago may no longer be relevant in today's context.

5.     Please conduct a quality assessment for each article included in the analysis using either the Newcastle Ottawa Scale or QUIPS methodologies, whichever is applicable.

6.     Kindly include the forest plot for the meta-analysis. If it is not possible, please provide a rationale for why only qualitative data presentation is feasible.

7.     In the discussion section, it would be valuable to reference the paper with the DOI 10.1111/ajco.13513, which explains the different barriers to successful implementation of HPV vaccination programs in low- and middle-income countries.

8.     What are the limitations of the present study? It would be helpful to discuss any limitations or potential biases that may affect the findings and conclusions.

9.     Lastly, please provide recommendations for future studies based on the findings of this research. Suggestions for further investigation and areas that require additional attention would be beneficial.

English is fine and requires minor editing.

Author Response

Reviewer 4: The study conducted an extensive analysis to explore the acceptability of the human papillomavirus (HPV) vaccine. Through a systematic review and meta-summary of qualitative research, the viewpoints of vaccine users, parents, and healthcare professionals were examined. A total of 32 articles were thoroughly analyzed. The research revealed three critical factors that significantly impact the acceptance of the vaccine: the availability of comprehensive information about the vaccine, concerns and apprehensions regarding potential side effects, and the association of the vaccine with sexuality. This review highlights the complex nature of acceptability and adherence to HPV vaccination, emphasizing the importance of understanding the perspectives of users, parents, and professionals for future interventions. While the study provided valuable insights, further revisions are suggested to improve the clarity and refinement of the manuscript.

  1. It is important to include the most recent epidemiological data available on HPV-related malignancies, as well as benign lesions such as skin warts.
    Response 1: Recent epidemiological data was included in the introduction. Please see the lines 33 to 38, and 42 to 43.
  2. Previous studies have explored the limitations of HPV vaccination accessibility and adherence. To differentiate this study from previously published literature, it is essential to refer to the following papers with their DOIs: 10.1093/ije/dyt049, 10.1016/j.vaccine.2020.07.055, 10.1016/j.eclinm.2021.100836.
    Response 2: The references suggested were added in the discussion to differentiate the aims from this paper (please see lines (199-200)
  3. There seems to be an issue with Figure 1. Please provide a detailed schematic diagram that illustrates the results of the search strategy.
    Response 3: The flowchart (Fig 1) was missing. We don’t understand what happened because the flowchart was in page 3 of the paper. In the new version the flowchart was sent as document also.
  4. Why was the search strategy limited only to February 2020? It is crucial to provide a rationale for this choice and consider including published literature beyond that timeframe to ensure the data reflects the current state of HPV vaccine uptake. Information from three years ago may no longer be relevant in today's context.
    Response 4: It should be noted that this situation was declared as a limitation of the study. The research project lasted 2 years, until the end of 2021. Given the pandemic situation, the research team decided not to include new references due to its possible effect on the results.
  5. Please conduct a quality assessment for each article included in the analysis using either the Newcastle Ottawa Scale or QUIPS methodologies, whichever is applicable.
    Response 5: The quality assessment for each article included in the analysis was done using Joanna Briggs Institute (JBI); the information about quality was probably not well explained in the first version of the article. Therefore, the original paragraph “The study applied the critical appraisal guideline of the Joanna Briggs Institute to each research [11], and none of the studies were excluded due to quality issues” was replaced by : "The quality of each article was evaluated with the specific guideline of JBI for qualitative studies[11]; it should be noted that after this evaluation, none selected article should have been eliminated." (please see lines 114 to 116)
  6. Kindly include the forest plot for the meta-analysis. If it is not possible, please provide a rationale for why only qualitative data presentation is feasible.
    Response 6: The forest plot was not included because the aim of this paper is related to the reasons that favor or do not favor the acceptability of HPV vaccines from a qualitative perspective. The reason to distinguish qualitative and quantitative methodologies was that the aims from each one is different. The qualitative studies (with a specific qualitative methodology) were about HPV vaccine acceptability and the quantitative studies were about effective educative interventions to increase the adherence. Because both aims are different and they are not comparable, it is not possible to include the forest plot from quantitative studies.
  7. In the discussion section, it would be valuable to reference the paper with the DOI 10.1111/ajco.13513, which explains the different barriers to successful implementation of HPV vaccination programs in low- and middle-income countries.
    Response 7: The paper suggested was included in the discussion section (please see lines 245 to 250)
  8. What are the limitations of the present study? It would be helpful to discuss any limitations or potential biases that may affect the findings and conclusions.
    Response 8: The principal limitation was described in the original version as: “Its major weakness is that the results were not published immediately after completing the analysis; therefore, evidence after 2020 was not considered”. In the new version the paragraph was modified “the principal limitation of this paper is that the evidence after 2020 was not considered, therefore the results are related to the dates included"
    A possible bias was included in the new version: “The possible bias of this study is the theoretical lens used in the first part of phase 1 of the analysis, since the findings were grouped according to the categories of the selected theoretical framework; however, this limitation is balanced in the second part of the same phase where through an inductive process, new categories were created”.(please see lines 202 to 205).
  9. Lastly, please provide recommendations for future studies based on the findings of this research. Suggestions for further investigation and areas that require additional attention would be beneficial.
    Response 9: Recommendations and suggestions were included in the new version. (please see lines 259 to 270)

Round 2

Reviewer 1 Report

For the "results" section, more scientific description should be provided to explain the study process, and how to interprete these results, which should lead to the conclusions.

Tables 4, 5 and 6 need better summary of the contents instead of listing everything, which greatly reduce the readability of the review. 

The conclusions from the current study is too broad and not very related to scientific aspects of vaccines. 

Overall, the logical connection between "results" and "conclusions" is weak. It is not clear how the contents in the results section lead to the conclusions, which seems subjective.

Author Response

  1. For the "results" section, more scientific description should be provided to explain the study process, and how to interprete these results, which should lead to the conclusions.

Response 1: With the objective of a more detailed description, a brief textual definition of each one of the stages of the process was added, extracted from the authors of the methodology used. Please the new lines added in each stage of the results section.  

  1. Tables 4, 5 and 6 need better summary of the contents instead of listing everything, which greatly reduce the readability of the review. 

Response 2: It is not possible to delete content, since it is the result of each of the stages described. Dividing lines were added to help the readability.

  1. The conclusion from the current study is too broad and not very related to scientific aspects of vaccines. 

Response 3: In our opinion, the conclusion is related to the results obtained in the qualitative synthesis process described by Sandelowski. For a better understanding, each theme found in the result was named in the discussion.

  1. Overall, the logical connection between "results" and "conclusions" is weak. It is not clear how the contents in the results section lead to the conclusions, which seems subjective.

Response 4: For a better understanding, the connection between result and conclusion was made explicit. Please see the new lines in the conclusion section.

Reviewer 4 Report

The study has been revised and is now acceptable for publication.

Author Response

Thank you 

Round 3

Reviewer 1 Report

For point #1, I don't think the added sentences in the Results section answer my question.

For point #2, can the authors summarize and make more concise tables with the contents? For a review, it is not necessary to directly quote the sentences from the original studies.

For point #3, I don't see the previous concern can be addressed with the current modification.

For point #4, the newly added lines (line 284-286) has many mistakes, for example:

“nine 9 qualitative dimensions” what does that mean? Please carefully check the manuscript.

“9 for parents and 4 for vaccinated indindividuals /adolescents)…”: there is a logical mistake with the previous sentence (as mentioned, the total is 9 dimensions, how can you separate them into two categories with 9 and 4?). Also, I did not see why “parents” and “vaccinated individuals” can be put together side by side as two same-level categories.

Minor editing of English language required.

Author Response

Response to review

  1. For point #1, I don't think the added sentences in the Results section answer my question.

Response: The suggestion was considered, and a more detailed description was provided to explain the results. Please see the new description highlighted in the results section. The figure 2 and Table 3 were improved with the new description also.

Table 4 was modified deleted information that was not important.

  1. For point #2, can the authors summarize and make more concise tables with the contents? For a review, it is not necessary to directly quote the sentences from the original studies.

Response: The suggestion was considered, and the tables were modified.

  1. For point #3, I don't see the previous concern can be addressed with the current modification.

Response: The conclusion was improved with a correct explanation about the origin of the conclusion and the importance of considers the qualitative studies as scientific evidence. Please see the new sentences added.

  1. For point #4, the newly added lines (line 284-286) has many mistakes, for example: “nine 9 qualitative dimensions” what does that mean? Please carefully check the manuscript. “9 for parents and 4 for vaccinated indindividuals /adolescents)…”: there is a logical mistake with the previous sentence (as mentioned, the total is 9 dimensions, how can you separate them into two categories with 9 and 4?). Also, I did not see why “parents” and “vaccinated individuals” can be put together side by side as two same-level categories.

Response: There was a mistake. The sentence was deleted.

Round 4

Reviewer 1 Report

I would like to thank the authors for the efforts in revising the manuscript. I don't have further suggestions for the manuscript.